# Implications of the Online Teaching Model Derived from the COVID-19 Lockdown Situation for Anxiety and Executive Functioning in Spanish Children and Adolescents

**DOI:** 10.3390/ijerph181910456

**Published:** 2021-10-05

**Authors:** Rocío Lavigne-Cervan, Borja Costa-López, Rocío Juárez-Ruiz de Mier, Marta Sánchez-Muñoz de León, Marta Real-Fernández, Ignasi Navarro-Soria

**Affiliations:** 1Department of Developmental and Educational Psychology, University of Malaga, 29071 Malaga, Spain; rlc@uma.es (R.L.-C.); rjrm@uma.es (R.J.-R.d.M.); 2Department of Health Psychology, University of Alacant, Ctra. Sant Vicent del Raspeig, s/n, 03690 San Vicente, Spain; borja.costa@ua.es; 3Neuropsipe, Child and Adolescent Neuroscience Center, 29010 Malaga, Spain; msanchez@neuropsipe.com; 4Department of Developmental and Educational Psychology, University of Alacant, Ctra. Sant Vicent del Raspeig, s/n, 03690 San Vicente, Spain; mrf44@alu.ua.es

**Keywords:** lockdown, COVID-19, anxiety, executive functioning, children, adolescents

## Abstract

Given the seriousness of the existing situation regarding the mental health of children and adolescents relating to the confinement period imposed due to COVID-19, we conducted this study to describe the effects of the confinement on state anxiety and executive functioning dimensions in a period of online educational modality. A sample of 953 children and adolescents was assessed. A sociodemographic questionnaire, the State Anxiety Inventory for Children (STAIC), and the Behavioral Evaluation of Executive Function (BRIEF-2) scale were applied. The analysis of the results indicates that 68.8% of children and adolescents presented medium–high levels of anxiety. Regarding sex, females showed higher levels of anxiety and worse levels of executive functioning. Although the group aged 11 to 18 years showed methodologically higher state anxiety (*p* = 0.041) than the group aged 6 to 10 years, the difference was not clinically relevant (δ = −0.113). The state anxiety variable was also correlated positively and significantly to the three executive functioning dimensions explored. In conclusion, it seems evident that COVID-19 lockdowns could have psychological and emotional effects on children and adolescents.

## 1. Introduction

At the end of 2019, the Wuhan Municipal Health and Sanitation Commission (China) produced a report indicating the detection of 27 cases of pneumonia with an unknown etiology. On 7 January 2020, the Chinese authorities identified a new type of virus from the Coronaviridae family, named SARS-CoV-2, as the causative agent of the pneumonia outbreak. Almost a month later, on 30 January 2020, the Emergency Committee of the International Health Regulations (IHR) declared a state of Public Health Emergency of International Importance (PHEIC) and renamed the infectious disease coronavirus 19 (hereafter, COVID-19). From that moment on, there was widespread concern in the media about the development of this new strain of coronavirus, until the World Health Organization (WHO), on March 11, elevated the situation caused by COVID-19 to the status of an international pandemic. Given the rapid spread of the virus and the serious consequences caused for a significant part of the world’s population, all countries have had to adopt measures to protect the health and safety of their citizens, contain the progression of the disease, and strengthen the public health system [1]. Social isolation, limitations on mobility, and mandatory quarantining of the population are some measures that have been immediately adopted to mitigate the impact of the virus at the health, social, and economic levels.

In Spain, on 14 March 2020, a state of alarm was decreed, and all levels of the educational system were forced to close. After that moment, people were subjected to more than a month and a half of confinement, without being able to leave their home, except for some essential activities. It is estimated that during March 2020, more than 3 billion people were confined worldwide. While the conditions of confinement have varied from country to country, this situation has affected the general population at different levels. Many families have seen their finances eroded, and businesses have gone bankrupt. On a socioemotional level, this situation has marked a turning point due to the great restrictions that have meant moving from more physical and close contact to a more technological and distant form of socializing. Few social events have had such an impact on human psychology as the COVID-19 pandemic, especially considering the number of people it has affected and the spheres of life it has influenced [2]. All of this has caused significant psychological stress in people from different countries [3,4], altering the mental health of adults, children, and adolescents and producing changes in mood, behavior, and daily habits [5,6,7,8,9,10,11]. 

The closure of educational centers immediately implied a total change in the teaching modality, and teachers, students, or parents were not prepared for it. The teaching modality changed from a face-to-face modality to an online modality at all educational levels, both synchronously and asynchronously, with all the difficulties that each entails. At the synchronous level, it was difficult to reconcile schedules and keep the children attentive through a screen on which their teacher appeared, giving class in real time. It was not easy at the deferral level either. Teachers had to share material, videos, and activities that students had to complete before a given date, leaving in the students’ hands (or rather, in their parents’) the challenge of planning and organizing all the activities entrusted to them, which became an arduous task that increased the levels of anxiety and stress derived from the situation even more. It should be noted that these measures have been implemented at all academic levels, regardless of the age or ability of the child. 

Classrooms were abruptly substituted by the different rooms of homes, the blackboard was replaced by the tablet or computer screen, the schoolyard by the living room or terrace, and physical and direct contact with classmates disappeared, along with extracurricular activities and outdoor games. Parents modified or duplicated their role. It was not enough to only be a parent; parents also had to become teachers, while teleworking and fulfilling with household responsibilities. In many cases, children and adolescents were required to have an autonomy that required a broad mastery of technology, which not all of them possessed, and some of them in fact achieved the opposite. Thus, most of them became dependent on the advice of adults (in most cases, parents), since without them, the execution and development of school tasks via the Internet would have been impossible. If we are talking about families without technological resources and without time to assist their children in these tasks, the situation became even more complicated. As a result, stress and anxiety levels increased, and the stressors were modified. Instead of worrying about having one’s name on the blackboard or the opinion of peers, the main stressor was the increased responsibility placed on the students, the total departure from people’s comfort zone, which was replaced by an environment not associated with academic purposes, the on-line environment, and the absence of authority figures and peers. 

Hence, anxiety states became prevalent during the confinement period among children and adolescents [10,12,13,14,15]. Several studies have shown that anxiety has a negative impact on the academic performance of children and adolescents [16,17] and on their levels of emotional control, which is especially important in situations, such as those experienced during these months due to the pandemic. 

Moreover, this circumstance may be crucial for some populations (e.g., individuals with autism or ADHD) [11,18,19] due to the special impact that it may have on different facets of their lives. Studies conducted in China [15,20,21,22,23], with a population aged 10.84 years on average, highlighted that 24.9%, 18.9%, 28.8%, and 37.4% of those assessed reported anxiety symptoms. Regarding the prevalence of anxiety problems as a function of sex, the data from the studies reviewed are inconclusive. Some results indicate a higher risk of anxiety in women [23,24], while others found no evidence to support this finding [21].

The National Child Traumatic Stress Network (2020) [25] states that the type of psychological response to the physical and social consequences of the occurrence of COVID-19 depends on the age of the subjects. Among children in early childhood education, the most frequent responses were manifestations of fear and reduced appetite, together with a notable increase in tantrums and complaints. At the same time, behaviors typical of an anxious attachment were be observed. In the primary education stage, between 6 and 12 years of age, irritability, disturbed sleep due to the presence of nightmares, lack of appetite, and disinterest in their peers or classmates, together with an exaggerated attachment to their parents, were more frequent. In the case of young people between 13 and 18 years of age, the symptoms were different. They presented a greater physical somatization, difficulties achieving regular sleep, and social isolation from their peers. In addition to all this, there was a decrease in energy, which was accompanied by a greater apathy and loss of interest in behaviors related to self-care and health. While this information has already been reported, it is essential to extend the research to shed further light on the psychological consequences of the response to the health measures adopted to palliate COVID-19, and in particular, information is needed on how it affects the child population, since these aspects of mental health have been studied extensively in university populations during lockdown, but less research has been conducted on populations of younger ages, which are clearly more vulnerable [14,20,26,27]. 

Along with emotional problems, it is evident that different cognitive processes are going to be affected by the sustained situation of unnatural human confinement. Several studies have shown interest in understanding how stress and anxiety are related to cognitive processes [28]. In particular, it has been observed that stress plays a significant role in the functioning of several brain areas, and some studies report a direct involvement in the frontal regions responsible for governing higher cognitive processes, such as executive functions. It has been shown that both chronic perceived stress and chronic stress impair functions such as planning, working memory, flexibility, and emotional management [29], showing a negative relationship between both dimensions [30,31]. Furthermore, during childhood and adolescence, brain plasticity plays a determining role in frontal and prefrontal areas, showing structural and functional variations in response to stress or behavioral anxiety [32,33].

The measures implemented to stop the advance of COVID-19 have meant, in the first place, confining the population on a massive scale. This type of measure has generated an important change in the routines of individuals, a reduction in face-to-face social relationships, frustration, boredom, and the perception of being alone. These circumstances have been very difficult to manage for many people [34]. Previous research focusing on the implications of lockdown periods for other health crises, such as severe acute respiratory syndrome (SARS) in 2003, Ebola in 2014, and influenza AH1N1 in 2009 and 2010, could help us to anticipate what psychological consequences could emerge from a situation like the current one [35]. The results of these studies highlight the prevalence of anxious symptoms in 20% of the study population and 18% in relation to depressive symptoms. In these processes, even higher scores than those of the ordinary population were detected in health system workers. Sprang and Silman [36] concluded from their research that children who suffered confinement during a period of health crisis were more likely to develop acute stress disorders, adjustment disorders, and grief. In fact, 30% of the child population studied, who suffered social isolation, had clinical symptoms of post-traumatic stress disorder. The CHINA-EPA-UNEPSA collaborative working group [37], in an investigation of the health crisis resulting from COVID-19, with a sample of 320 children and adolescents in Shaanxi province, indicated, as the most significant results of their study, that children aged 3 to 6 years were more likely than older children to show symptoms of ambivalent anxious attachment and fear that their family might become infected (*p* = 0.002). In addition, children and young people aged 6 to 18 years had a higher incidence of attentional deficits (*p* = 0.003). Attachment difficulties, inattention, and irritability were the most serious psychological consequences presented by children and adolescents of all age groups. In fact, these elevated scores for fear, anxiety, and other adverse emotional states were higher in subjects living in areas where the epidemic reached more significant proportions. As a general conclusion of all the studies cited, it can be stated that after the quarantine period, behavioral changes are maintained in the research participants, such as the maintenance of alertness and obsessions, such as excessive hand washing or avoidance of crowded environments. This information indicates that there is evidence that there are groups that have suffered and will suffer to a greater degree from measures such as social isolation, including those with a previous mental pathology, pregnant women, the elderly, health care workers, children, and adolescents [38]. 

Given the aforementioned facts, and given the severity of the situation that exists with regard to the mental health of children and adolescents, this study had the following objectives: (1) to describe the prevalence of state anxiety in children and adolescents in an educational modality period of non-attendance due to the COVID-19 lockdown; (2) to examine whether there are significant differences in state anxiety by gender and age group; (3) to analyze the effects of COVID-19 confinement on dimensions of executive functioning in children and adolescents (emotional self-regulation, cognitive flexibility, and planning/organization); (4) to examine the relationship between dimensions of executive functioning and state anxiety in children and adolescents; (5) to analyze sex differences in executive functioning; and (6) to find out differences in state anxiety between sexes by level of executive functioning.

## 2. Materials and Methods

### 2.1. Study Design and Participants

In this cross-sectional observational study, we included 953 children and adolescents. The sample had almost the same number of people of both biological sexes, most of which were Spanish, with a mean economic income above 950 euros per month. The characteristics of the participants are summarized in Table 1. 

### 2.2. Variables and Instruments

To conduct the present research, an instrument was designed to be filled out by family members (fathers, mothers, and/or legal guardians) of the participants and by the participants themselves from 6 to 18 years of age, asking for help—if necessary—from the adults for the proper fulfillment of the instrument. The time required to complete it was roughly 15–20 min. The instrument was structured in three parts. The first part was an ad hoc questionnaire aimed at collecting sociodemographic data and data related to the health status of the participants. The second was on the level of state anxiety, and the third was on the level of executive functioning. 


*Part 1. Ad hoc sociodemographic and clinical data questionnaire.*


The ad hoc sociodemographic questionnaire designed for the research included variables such as the schooling level of the persons who filled it out, the monthly income of the family, whether they were working or unemployed before and after the declaration of the state of alarm, the type of work they do, the size of the residence in which they live during the quarantine, and whether it has terraces, gardens, balconies, or something similar. In addition, information was collected on whether the children or adolescents have any medical and/or psychological diagnosis, as well as whether they receive pharmacological treatment or psychotherapy. Finally, information was gathered on whether there were people in the family unit with risk factors for contracting the coronavirus and whether any member had tested positive in a COVID-19 test.


*Part 2. Spielberger’s State-Trait Anxiety Self-Assessment Questionnaire in Children (State Anxiety Inventory for Children, STAIC) *[39]*.*


This instrument is based on the State-Trait Anxiety Inventory (STAI), which aims to assess both state anxiety and trait anxiety by means of two independent scales comprising 10 items each. Each item provides 3 response options (1 = not at all, 2 = somewhat, 3 = very much). In particular, for this study, we used the scores obtained on the state anxiety scale, which provided information on how the participant is feeling at a given moment, such as the one he/she was experiencing during the lockdown. It is an instrument that presents a high reliability through the Kuder Richardson KR-20 internal consistency coefficient of 0.91 for State Anxiety [39]. While the original application of the instrument was self-administered, for the study, as mentioned before, it was pointed out that if the wording of the items was difficult to comprehend, especially for younger children, the participants should fill them out with the help of adults (fathers, mothers, and/or legal guardians).


*Part 3. Behavioral Evaluation of Executive Function: BRIEF-2 *[40]*. Spanish adaptation.*


This was a questionnaire to obtain executive functioning information in the environment in which individuals from 5 to 18 years of age live in their daily lives. It consists of 63 items, with three possible response options each (always, sometimes, or never). These items were distributed in nine scales that provide measures of inhibition, self-control, cognitive flexibility, emotional management, initiative, working memory, planning and organization, task supervision, and organization of materials [40]. These domains were grouped into three indices: a first index of behavioral regulation, a second of emotional regulation, and a third of cognitive regulation. The global index of executive functioning could also be acquired. It comprised two versions, one for the family and the other for the school, both with the same structure. 

As the original instrument was very extensive and the participants were confined, it was decided to use a brief screening for the present study by selecting 12 items from the family version that were directly related to the objectives of the study. For this purpose, 6 of the 8 items of the emotional self-regulation subscale (AR), 4 of the 8 items of the cognitive flexibility subscale (FC), and 2 of the 8 items of the planning and organization subscale (PO) were selected. Due to the selection of items for this study, the discrimination indices of these items are presented as indicators of the test–retest reliability for each of the subscales. Specifically, the items comprising the emotional control subscale had a well-defined discrimination index ranging from 0.46 to 0.72. The items selected from the flexibility subscale also had a good quality of discrimination between 0.29 and 0.37. Additionally, something similar occurs in the items selected from the planning and organization subscale, where there was an excellent discrimination of 0.56 in both of them. 

### 2.3. Procedure

A single assessment was carried out by means of an online questionnaire that combined all the instruments described above. This format was chosen because of the situation of confinement decreed in the country. For convenience, the sample was selected on the basis of age. 

On Thursday, 16 April 2020, at 19.30 h, the questionnaire was sent out and disseminated through social networks and local media, requesting citizen collaboration to participate in the study. The questionnaire was online for two weeks. 

At the beginning of the completion of the instrument, informed consent was required from the fathers, mothers, and/or legal guardians of the participants, who were informed of the anonymity of their data and the exclusive use to which it would be put. At the same time, the ethical principles of the Declaration of Helsinki (World Medical Association, 2013) were respected. This study was carried out in accordance with the Ethics Committee and the Vice-rectorate for Research and Knowledge Transfer of the University of Alicante (UA-2020-05-12), approved on 16 July 2020.

### 2.4. Data Analysis

Statistical analysis started with a database refinement in order to remove duplicates. An exploratory analysis was conducted. 

Following a quantitative methodology, descriptive analyses (mean and standard deviation) of the sociodemographic characteristics (age, sex, nationality, and income level) of the sample were carried out. The frequencies, percentages, and descriptive analyses (means and standard deviation) of the state anxiety and executive functioning domains (emotional self-regulation, cognitive flexibility, and planning/organization) were also calculated. Pearson’s correlations among state anxiety and executive functioning domains were analyzed and interpretated according to Hernández–Lalinde et al. [41].

Moreover, the assumptions of normality and homoscedasticity were respectively checked with Kolmogorov–Smirnov and Levene tests [42]. Differences in the state anxiety between primary and secondary school individuals were identified by employing Student’s *t*-test. Moreover, differences in executive functioning between sexes were analyzed with Student’s *t*-test, and state anxiety according to the level of executive functioning was evaluated by employing the one-factor ANOVA test. Furthermore, *t*-test was applied to find differences in state anxiety between sexes depending on the level of executive functioning. *p* < 0.05 was considered significant in all cases. 

Finally, Cohen’s delta (δ) and Hays’ omega (ω²) were calculated to find out the clinical impact on the population of the results obtained. ω² was selected as the most suitable effect size’s indicator due to it was not influenced by the sample size [43]. The frequencies, percentages, and descriptive analyses were analyzed using the SPSS Statistical software package, IBM SPSS Statistics v26 (Armonk, NY, USA). Pearson’s correlations and *t*- and ANOVA tests were conducted using the R package [44,45].

## 3. Results

First, with the objective of describing the prevalence of state anxiety in children and adolescents in a period of the non-attendance educational modality due to the lockdown caused by COVID-19, we carried out a descriptive analysis, in which the highest percentage of the sample reached high state anxiety scores. Indeed, 69% of children and adolescents presented medium or high anxiety (Figure 1). 

Besides, the violin box plot represents different densities in low, medium, and high state anxiety. Boys indicate higher percentages in low (31.8%; *n* = 163) and medium anxiety (34.6%; *n* = 177). However, the percentage of girls is higher in high anxiety (39.9%; *n* = 175) (Figure 2). 

Likewise, to analyze the differences in state anxiety between age groups, a Student’s *t*-test was carried out. As can be observed in Table 2, regarding state anxiety during the lockdown caused by COVID-19, differences were found between age groups of children and adolescents. In this sense, individuals from 11 to 18 years old present a higher state anxiety (Table 2). However, the calculated effect size was small.

Additionally, to analyze the effects of the COVID-19 lockdown on executive functioning dimensions in children and adolescents (emotional self-regulation, cognitive flexibility, and planning/organization) in a period of the non-attendance educational modality due to the lockdown caused by COVID-19, we conducted a descriptive analysis. According to the means of all three domains, children and adolescents seemed to present a good executive functioning (ESR = 21.86 ± 10.007; CF = 13.49 ± 6.553; PO = 6.49 ± 3.165). However, as the box plot showed (Figure 3), the density of the sample was distributed beyond the mean. While all three EF dimensions presented a tendency toward low scores, the emotional self-regulation domain scores might be almost homogeneous. Similarly, in the cognitive flexibility domain, the children and adolescents scores could be tendentiously homogeneous. Additionally, in the planning and organization domain, the scores were clearly above the mean, considering that the maximum score was 14.

Regarding the examination of the relationship among executive functioning dimensions and state anxiety in children and adolescents, Pearson’s correlations were calculated. The individuals show a positive, strong, and significant correlation with emotional self-dysregulation (0.529), a positive, moderate, and significant correlation with cognitive rigidity (0.429), and a positive, weak, but significant correlation with non-planning/organization (0.250) (Figure 4). 

A *t*-test was applied to analyze differences in executive functioning between sexes, and one-factor ANOVA was carried out to find out differences in state anxiety among executive functioning levels. Thus, significant differences were identified between the sexes of the participants in executive functioning (t[9 49] = 5.04; *p* < 0.001; δ = 0.328). Moreover, differences in state anxiety were found among the levels of executive functioning (F[2] = 170; *p* < 0.001; ω² = 0.262). As indicated in Table 3 and Figure 5, *t*-test analysis was conducted to examine the differences in state anxiety between the sexes of the participants in each level of executive functioning. Significant differences in state anxiety were found between: (a) men with good, moderate, and bad executive functioning; (b) men with good EF and women with moderate and bad EF; (c) women with good, moderate, and bad EF; (d) women with good EF and men with bad EF; (e) men with moderate EF and women with moderate and bad EF; (f) men with moderate and bad EF; and (g) women with moderate and bad EF and men with bad EF.

## 4. Discussion

During the last year, the study of the effects of the COVID-19 pandemic on different variables aroused great interest in the scientific community, which could not be otherwise. It is true that most studies either include the entire population in their samples or carry out research with specific populations of adults especially affected by the disease or its consequences [14,15,20,26,27,46,47,48,49,50], although more and more studies are appearing with children and adolescents, both neurotypical and with neurodevelopmental disorders.

Evidently children and adolescents are not indifferent to the dramatic impact of the COVID-19 epidemic. The change in their routines, to which they have been abruptly subjected, concerns us and prompts us to try to understand the state of their mental health and how they are affected by the changes in their social interaction, the educational centers, and the teaching model. Early research at the international level has shown that as a result of these changes in daily routine, a high prevalence of anxiety states has been observed during the period of confinement among children and adolescents [12,13,14]. The data obtained from the analysis of the results of the first objective are in line with those found in the cited studies, indicating that among a sample of 953 children and adolescents (aged 6 to 16 years, with a mean age of 10.84 years), 31% present low levels of state anxiety, 32.4% present medium levels, and 36.4% present high levels. A detailed analysis by sex shows that males show higher percentages of low (31.8%; n = 163) and medium (34.6%; n = 177) anxiety, unlike females, of which approximately 40% (n = 175) show high anxiety. Research data studying sex differences are not conclusive. Some point to a higher risk of anxiety in the female population [23,24], while others find no evidence of this [21].

In relation to age groups, we also found that the group aged 11 to 18 years showed a greater state anxiety (*p* = 0.041; d = −0.113) than the group aged 6 to 10 years. In addition, Martínez et al. (2020), by means of an online survey during the confinement of 425 Spanish children and adolescents aged 8 to 17 years, found some elements to be potential sources of discomfort generating state anxiety. Specifically, they pointed out that the respondents felt fear (16.2%), sadness (28%), worry (36.7%), and boredom (61.6%). They found that between sex and age groups, there were interesting differences to be taken into account. In girls, the feelings were more acute than in boys, with the exception of boredom, which was better tolerated by girls. As for the age groups (8–12 years and 13–17 years), the same was true for the older ones as for the girls, with the exception of worry, for which the older ones obtained very high scores. All this seemed to indicate that the older the age, the greater the accuracy of the knowledge of the complexity of the social world around them, and the greater the awareness of their fragility in the context of the pandemic.

On the other hand, Jiao et al. [37], using a sample of 320 children and adolescents aged 3 to 18 years, with the aim of detecting behavioral and emotional disorders during confinement, found that the most common problems that they presented were related to attention, irritability, fear of asking about the coronavirus, and ambivalent attachment behaviors, especially behaviors that are correlated with the state anxiety variable. Similarly, another study conducted in China by Cao et al. [20] with university students during the beginning of the pandemic found that 24.9% had severe–moderate to medium anxiety (0.9%, 2.7%, and 21.3%). They also found that social support was correlated negatively with the level of anxiety (*p* < 0.001), as social support, due to the circumstances and measures proposed to prevent the spread of COVID-19, was prevented. These conditions, which generate social isolation, were configured as a non-normative stressor that increases the possibility of presenting mental problems for the first time or the exacerbation and recurrence of pre-existing mental disorders [51]. There is no doubt that anxiety is considered a mental disorder that can affect quality of life, negatively impacting aspects, such as emotional states, cognitive functioning, or psychosocial adaptation [52].

To determine the effects of the confinement to which we have been subjected on some domains of executive functioning, we found that in emotional control, cognitive flexibility, and planning/organization, the studied children and adolescents showed average values (ESR = 21.86 ± 10.007; CF = 13.49 ± 6.553; PO = 6.49 ± 3.165). However, through a detailed analysis of the data and through an interpretation of the box plot graph, we found that 48% of the participants were above average in emotional control, 42% were above average in cognitive flexibility, and 37% were above average in the ability to plan and organize. All this shows that the situation experienced is making a dent not only in the levels of state anxiety, but also at the executive level in the dimensions studied.

In this sense, just as there is a large body of work studying the possible effects of confinement on emotional variables, such as anxiety [5,46,53,54,55], there are few studies that explore the impact on cognitive variables, and even fewer on executive functions [11,56]. Thus, one of the few studies carried out to date has been conducted by Lavigne et al. [56], demonstrating that there are positive correlations between the state of anxiety and alterations in executive functioning. In addition, they proved, through a path analysis, that state anxiety plays a leading role in explaining the alteration of executive functioning. Furthermore, if neurological difficulties prior to the confinement situation are added to these circumstances described above, another recent study [11] determines that in comparison with the neurotypical population, the subjects who present difficulties obtain higher anxiety scores and a significant worsening of their executive functioning.

Langarita and Gracia [57] conducted a systematic review, in which they analyzed forty articles, gathering a total sample of 1098 cases of subjects with symptoms of generalized anxiety. They concluded, from their data, that elevated levels of anxiety are correlated with decreased executive functions, obtaining significantly lower scores with respect to a control group in cognitive flexibility [58], negative valence emotion regulation [59], and cognitive inhibition [60]. The correlation analyses on the state anxiety variable performed for this study showed positive and significant correlations in the three executive dimensions explored. The correlations found were strong, moderate, and weak in relation to emotional self-regulation (0.529), cognitive rigidity (0.429), and planning/organizing ability (0.250), respectively. The cognitive processes of children and adolescents are influenced by their emotions. High levels of stress can cause alterations in the prefrontal lobes, resulting in dysfunctions at the executive level [61].

Likewise, the last objective completes this work by confirming once again the role played by anxiety in executive functioning. Thus, it is observed that higher levels of anxiety worsen executive functioning in the sample of children and adolescents. In addition, the analyses of the present study show that it is the girls who present higher levels of anxiety and, thereby, worse executive functioning. In fact, Figure 5 shows an almost exponential deterioration of executive functions, which could be directly related to state anxiety.

It is evident that both the state of anxiety and the alterations in executive functioning can vary over time and fluctuate in intensity, especially in a pandemic situation and with the social restrictions to which we have all been subjected. This is particularly important for children and adolescents. Psychological reactions to a pandemic are usually severe, and it is possible to observe long-term emotional sequelae [9]. The quarantine situation has caused parents and children to sharpen their wits in an attempt to adapt and “survive” by making the time pass in an optimal, friendly, and fun way. Even so, social limitations have caused behavioral and emotional changes in children and adolescents, some more intensely than others. While they continue to have online contact with their peers, teachers, relatives, etc., and while adults have been trying to minimize the effects of the life changes that children and adolescents have been forced to undergo, the conditions are not the same as before and are not the most optimal for stable socioemotional development, so the effects on mental health are and will be increasingly evident [5,6,7,8,9,11,56,61,62].

### 4.1. Limitations

While the present study advances our understanding of the psychological effects of the pandemic in children and adolescents, some limitations should be considered. Despite the high number of participants, the assessment of the study was online, so a number of samples were lost due to wrong answers and misunderstanding of the test. Moreover, there was no measure before the confinement, so we cannot make a comparison of the results. Finally, the prevalence of depression and suicidal thinking, which has been recently associated with the confinement in children and adolescents, was not evaluated in this study.

### 4.2. Future Research

Based on this study, two future research projects are planned, which will complement the results obtained. Firstly, new information will be collected from the same sample, once the current situation has been overcome, and the study subjects have recovered part of the psychosocial habits that characterize their daily lives. Finally, intervention programs will be developed to prevent and treat the difficulties found in the subjects surveyed, with the aim of minimizing as much as possible the possible consequences that confinement may have generated, both in the subjects’ anxiety levels and in their executive functioning.

## 5. Conclusions

In conclusion, our study highlights the prevalence of anxiety and a bad executive functioning in children and adolescents during the online learning situation resulting from the pandemic. Individuals who suffer high levels of anxiety tend to present a worse emotional management, cognitive rigidity, and disorganization. If these psychological effects persist over time, school performance could be affected, since these variables have a direct impact on it. Focusing on this population provides information that can be used by educational experts and clinicians to establish prevention and intervention strategies to mitigate the deterioration of mental health and its negative consequences on students’ learning. Taking care of mental health is currently one of the most relevant factors in childhood and adolescence.

This work emphasizes the importance of early intervention with children and adolescents in order to be able to detect different problems that can trigger this type of situation, and contributing to the improvement of a general well-being, which reaches a great relevance for human development.

## Figures and Tables

**Figure 1 ijerph-18-10456-f001:**
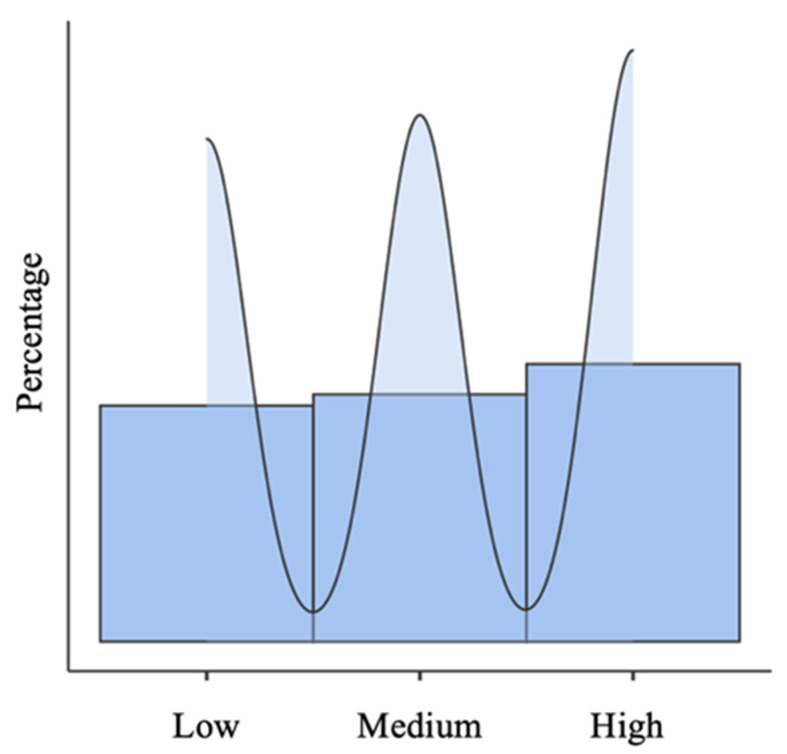
Percentage of the prevalence of state anxiety in children and adolescents (*n* = 953).

**Figure 2 ijerph-18-10456-f002:**
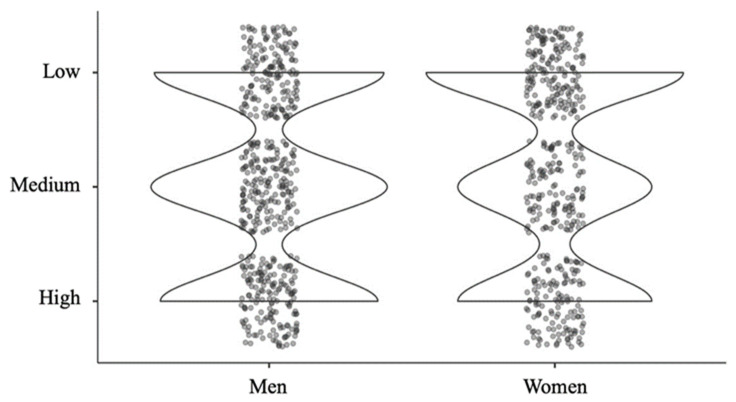
Violin box plot. Low, medium, and high state anxiety represented by the density of the sample and separated by sex.

**Figure 3 ijerph-18-10456-f003:**
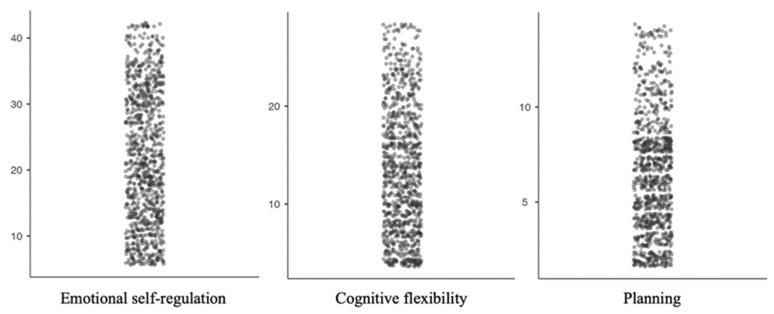
Box plot. Distribution of the sample in the emotional self-regulation, cognitive flexibility, and planning domains of executive functioning.

**Figure 4 ijerph-18-10456-f004:**
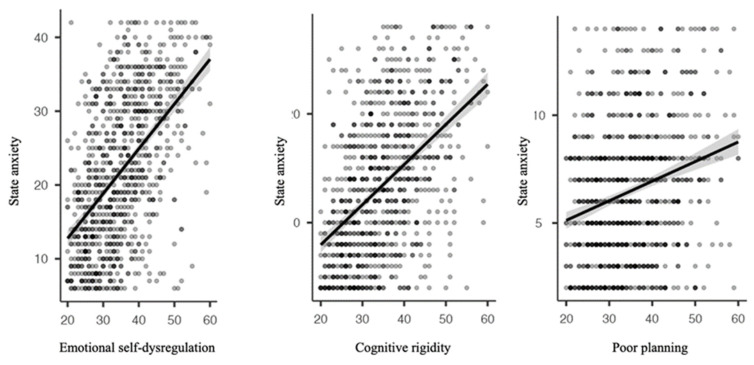
Correlations between state anxiety and each of the executive functioning dimensions.

**Figure 5 ijerph-18-10456-f005:**
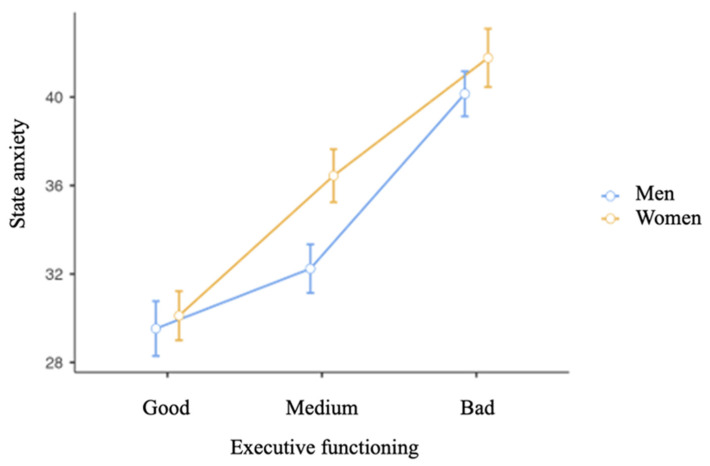
Means of state anxiety between the sexes of the participants in each level of executive functioning.

**Table 1 ijerph-18-10456-t001:** Sociodemographic data of the participants.

		N = 953
**Age**		10.84 ± 3.286
	Men	10.94 ± 3.222
	Women	10.72 ± 3.350
**Sex**		
	Men	512 (53.8%)
	Women	439 (46.2%)
**Nationality**		
	Spanish	914 (95.9%)
	Other	39 (4.1%)
**Income level**		
	Low (<EUR 950)	219 (23.0%)
	Average (=EUR 950)	141 (14.8%)
	High (>EUR 950)	593 (62.2%)

**Table 2 ijerph-18-10456-t002:** Differences in state anxiety between age groups.

Age Group (Years)	Mean (SD)	t(df)	*p*	Effect Size
6–10	34.5 (±8.26)	t (951) = −1.74	0.041	δ = –0.113
11–18	35.5 (±9.14)			

df (degrees of freedom).

**Table 3 ijerph-18-10456-t003:** A *t**-test analysis*. Differences in state anxiety between sexes, according to the level of executive functioning.

		t(df)	*p*	Effect Size
Men with Good EF	Women with Good EF	t(945) = −0.690	0.983	d = −0.079
	Men with Moderate EF	t(945) = −3.212	0.017 **	d = −0.367
	Women with Moderate EF	t(945) = −7.881	<0.001 ***	d = −0.936
	Men with Bad EF	t(945) = −12.973	<0.001 ***	d = −1.437
	Women with Bad EF	t(945) = −13.284	<0.001 ***	d = −1.657
Women with Good EF	Men with Moderate EF	t(945) = −2.672	0.082	d = 0.288
	Women with Moderate EF	t(945) = −7.617	<0.001 ***	d = −0.857
	Men with Bad EF	t(945) = −13.055	<0.001 ***	d = 1.358
	Women with Bad EF	t(945) = −13.285	<0.001 ***	d = −1.578
Men with Moderate EF	Women with Moderate EF	t(945) = −5.079	<0.001 ***	d = −0.569
	Men with Bad EF	t(945) = −10.337	<0.001 ***	d = −1.070
	Women with Bad EF	t(945) = −10.901	<0.001 ***	d = −1.290
Women with Moderate EF	Men with Bad EF	t(945) = −4.602	<0.001 ***	d = 0.501
	Women with Bad EF	t(945) = −5.877	<0.001 ***	d = −0.721
Men with Bad EF	Women with Bad EF	t(945) = −1.912	0.395	d = −0.220

df (degrees of freedom). ** *p*<0.01; *** *p*<0.0001.

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
