# Peer review of "Implications of the Online Teaching Model Derived from the COVID-19 Lockdown Situation for Anxiety and Executive Functioning in Spanish Children and Adolescents"

_ijerph, 2021, doi:10.3390/ijerph181910456_

Round 1

Reviewer 1 Report

Many sources in the literature are older than ten years. I encourage authors to consider whether including these resources in an article is really important. Also, in the bibliography, the date is highlighted in bold somewhere and not elsewhere. It needs to be unified.

Within the methodology, it should be clear how the respondents were selected. It was a random selection, how this selection took place, etc.

The abstract states that "the group aged 11 to 18 years showed significantly higher state anxiety (p = 0.041) than the group aged 6 to 10 years". The p-level value in this case is borderline and due to the number of respondents, statistical significance is expected. At this point, I consider it appropriate to add effect size in addition to statistical significance. The value of "d" can be found in the chapter "results" and how the authors themselves add "The calculated effect size is small". This mention completely reverses the meaning of the text from the abstract.

The methodology includes: "To conduct the present research, an instrument was designed". If the instrument was not standardized, then I lack a description of its psychometric properties, etc.

In the "Data Analysis" section, I would expect a more detailed description of what data can be obtained from the tools. Are they always continuous? Metric? Ordinal? How was normality handled? Why do parametric tests count straight away?

Why was ANOVA performed when only two groups were compared? For ANOVA, in addition to Hays's omega, it is possible to use Fisher's eta. Justify the use of Hays omega.

What test was used for the post-hoc analysis? According to Table 3, it seems that the repeated t-test? Why wasn't the most well-known Tukey's test (or Games-Howell, if no agreement of variance is reached) or another of the tests such as Schefény, etc. used? In addition, in Table 3 I lack the justification for the significance of the values ​​"p" and "d", for example by means of the symbol *, **, ***.

The discussion part is largely a recapitulation and the conclusion is very short. My next complaint is related to the fact that the authors themselves describe: "Moreover, there was no measure before the confinement, so we can not make a comparison of the results." Although it is expected that the results are largely affected by the covid-19 pandemic, the authors can neither confirm nor deny this fact on the basis of the data.

Author Response

Dear Reviewer,

Thank you for accepting to review the article, as well as for the time and effort spent both in reading it and making suggestions. Your contributions are particularly interesting and allow for the improvement of the document. In the following, each and every question is answered, point by point, and the specific line of the text where you can find the modifications is indicated.

Point 1: Many sources in the literature are older than ten years. I encourage authors to consider whether including these resources in an article is really important. Also, in the bibliography, the date is highlighted in bold somewhere and not elsewhere. It needs to be unified.

Following your suggestions, we have proceeded to eliminate those references older than ten years. Specifically, references 29, 30, 32, 35 and 56 have been removed. Likewise, the format of the publication date of the bibliographical references has been unified. 

Point 2:  Within the methodology, it should be clear how the respondents were selected. It was a random selection, how this selection took place, etc.

In point 2 (Materials and Method), in section 2.3. (Procedure), it is explained that due to the situation of confinement decreed in the country, a convenience sample was selected, based on the age of the respondents. 

The questionnaire, designed as explained in section 2.2. (variables and instruments), was sent online in mid-April 2020, making use of social networks and local media, and remained active for two weeks. As participants received the questionnaire, they filled it in. If they met the age range required for inclusion in the study, the data were incorporated into our database, so that they could be exploited at a later date.

Point 3: The abstract states that "the group aged 11 to 18 years showed significantly higher state anxiety (p = 0.041) than the group aged 6 to 10 years". The p-level value in this case is borderline and due to the number of respondents, statistical significance is expected. At this point, I consider it appropriate to add effect size in addition to statistical significance. The value of "d" can be found in the chapter "results" and how the authors themselves add "The calculated effect size is small". This mention completely reverses the meaning of the text from the abstract.

We have added the effect size’s value to the abstract. According to Field (2009), differences are significant, but regarding your comment, we have changed the text. Differences are methodologically significant, but they might not be clinically relevant. Maybe, we should increase the sample size to find out whether these differences carry on actually being relevant to the young population. 

Point 4: The methodology includes: "To conduct the present research, an instrument was designed". If the instrument was not standardized, then I lack a description of its psychometric properties, etc.

There has probably been a misunderstanding in this section. We clarify it below so that there is no confusion, as two standardised instruments were used to collect the data: 

Spielberger's State Anxiety Inventory for Children (STAIC);

Behavioural assessment of executive function: BRIEF-2 (Spanish version).

Specifically, the instrument was designed online due to the particular situation we had to go through. For this purpose, it was structured in three parts. The first part consisted of an ad hoc questionnaire designed to collect socio-demographic data and data related to the health status of the participants. The second part consisted of Spielberger's Trait-State Anxiety Self-Assessment Questionnaire in Children (State Anxiety Inventory for Children, STAIC) to measure the level of state anxiety. And the third, by the Spanish version of the Behavioral Evaluation of Executive Function: BRIEF-2, to measure the level of emotional self-regulation, cognitive flexibility and planning and organisation. 

In relation to the latter instrument, given that the original instrument is very extensive and the participants were confined, it was decided to use a brief screening for the present study by selecting 12 items from the family version that were directly related to the objectives of the study. For this purpose, 6 of the 8 items from the emotional self-regulation (AR) subscale, 4 of the 8 items from the cognitive flexibility (CF) subscale and 2 of the 8 items from the planning and organisation (PO) subscale were selected. Due to the selection of items for this study, the discrimination indices of these items are presented as indicators of test-retest reliability for each of the subscales. Specifically, the items comprising the emotional control subscale have a well-defined discrimination index ranging between 0.46 and 0.72. The items selected from the flexibility subscale also have a good discrimination quality between .29 and .37. In addition, something similar occurs in the selected items of the planning and organisation subscale, where there is an excellent discrimination of .56 in both. 

All this information has been clarified in section 2.2: variables and instruments.

Point 5: In the "Data Analysis" section, I would expect a more detailed description of what data can be obtained from the tools. Are they always continuous? Metric? Ordinal? How was normality handled? Why do parametric tests count straight away?

Thank you for this comment. Data analysis section was modified and completed with further information (lines 262-280). Statistical assumptions (normality and homoscedasticity) were verified in order to choose the most suitable variance test. p-values which were above 0.05 were assumed to determine that parametric tests were available for this study. 

Point 6: Why was ANOVA performed when only two groups were compared? For ANOVA, in addition to Hays's omega, it is possible to use Fisher's eta. Justify the use of Hays omega.

Thank you for this appreciation. We have changed the tests applied before for a further comprehension of our results. Thus, T-test has been applied to find out sex differences in executive functioning. Also, T-test has been used to analyze sex differences in state anxiety accoding to the level o executive functioning. Finally, an one-factor ANOVA was used to analyze the differences in state anxiety among executive functioning levels. 

On the other hand, Hays’ Omega was used since it was the most suitable effect size’s indicator. It was not influenced by the sample size, as we have explained in the text (see Data analysis section and also lines 341-348). 

Point 7: What test was used for the post-hoc analysis? According to Table 3, it seems that the repeated t-test? Why wasn't the most well-known Tukey's test (or Games-Howell, if no agreement of variance is reached) or another of the tests such as Schefény, etc. used? In addition, in Table 3 I lack the justification for the significance of the values ​​"p" and "d", for example by means of the symbol *, **, ***.

As we have said before, we have changed the analysis tests. Therefore, there is no post-hoc analysis. We replaced them with Student's T-test and the text has been modified too. 

Moreover, in Table 3, we have added the symbols you mentioned such as **, ***. 

Point 8: The discussion part is largely a recapitulation and the conclusion is very short. My next complaint is related to the fact that the authors themselves describe: "Moreover, there was no measure before the confinement, so we can not make a comparison of the results." Although it is expected that the results are largely affected by the covid-19 pandemic, the authors can neither confirm nor deny this fact on the basis of the data.

In response to your suggestions, we proceed to revise the discussion and expand on the conclusions (see changes in the text, sections 4 and 5).

In relation to the last observation made, it is worth mentioning that scores prior to the COVID-19 were not obtained due to the obviousness of the unexpected situation that occurred. Thus, we proceeded to compare means found in each of the variables analysed with the means of the scales published for each instrument used, as well as the studies of their psychometric properties. The results obtained are very interesting, as can be seen in section 3. 

Reviewer 2 Report

It would be useful to have an image and a better description of the items of the questionnaire created ad hoc.

The STAIC test could be completed with the help of an adult but this could affect the type of responses; it could be useful to know the percentage of subjects who were helped.

You have made a selection of items in BRIEF 2, may you specify which ones you used?

It should be better explained how the association between levels of state anxiety, executive functions and the online teaching model took place; did you assess anxiety levels and executive functions even in the period preceding the covid?

In the discussion you said that "These studies do not specifically take into account the most vulnerable population that will also constitute the future of our society, a group to which particular importance must be attributed, namely children and adolescents" but there are studies that evaluate the impact of the pandemic on children, for example in the study by Sergi et al. The "Autism, Therapy and COVID-19" group is made up of children with ASD, so it would be better to expand the literature.

Author Response

Dear Reviewer,

Thank you for accepting to review the article, as well as for the time and effort spent both in reading it and making suggestions. Your contributions are particularly interesting and allow for the improvement of the document. In the following, each and every question is answered, point by point, and the specific line of the text where you can find the modifications is indicated.

Point 1: It would be useful to have an image and a better description of the items of the questionnaire created ad hoc. 

The questionnaire related to sociodemographic and clinical data has been created ad-hoc. The ad hoc sociodemographic questionnaire designed for the research included variables such as the schooling level of the persons who filled it out, the monthly income of the family, whether they were working or unemployed before and after the declaration of the state of alarm, the type of work they do, the size of the residence in which they live during the quarantine, and whether it has terraces, gardens, balconies, or something similar. In addition, information was collected on whether the children or adolescents have any medical and/or psychological diagnosis, as well as whether they receive pharmacological treatment or psychotherapy. Finally, information was gathered on whether there were people in the family unit with risk factors for contracting the coronavirus and whether any member had tested positive in a COVID-19 test.  (See section 2.2., variables and instruments). 

For the specific questionnaire, we provide you with the link where you can access it and review its structure: Confinamiento por COVID-19: impacto psicológico en niños y adolescentes (google.com)

Point 2: The STAIC test could be completed with the help of an adult but this could affect the type of responses; it could be useful to know the percentage of subjects who were helped.

In relation to explaining that they could count on the support of adults to fill in the questionnaire, we meant that only if they did not understand a word of an item well, could they be justified in asking for help.

Point 3: You have made a selection of items in BRIEF 2, may you specify which ones you used?

Behavioral Evaluation of Executive Function (BRIEF-2), was a questionnaire to obtain executive functioning information in the environment in which individuals from 5 to 18 years of age live in their daily lives. It consists of 63 items, with three possible response options each (always, sometimes, or never). These items are distributed in nine scales that provide measures of inhibition, self-control, cognitive flexibility, emotional management, initiative, working memory, planning and organization, task supervision, and organization of materials. These domains are grouped into three indices: a first index of behavioral regulation, a second of emotional regulation, and a third of cognitive regulation. The global index of executive functioning can also be acquired. It comprises two versions, one for the family and the other for the school, both with the same structure. As the original instrument is very extensive and the participants were confined, it was decided to use a brief screening for the present study by selecting 12 items from the family version that were directly related to the objectives of the study. For this purpose, 6 of the 8 items of the emotional self-regulation subscale (AR), 4 of the 8 items of the cognitive flexibility subscale (FC), and 2 of the 8 items of the planning and organization subscale (PO) were selected. Due to the selection of items for this study, the discrimination indices of these items are presented as indicators of the test–retest reliability for each of the subscales. Specifically, the items comprising the emotional control subscale have a well-defined discrimination index ranging from .46 to .72. The items selected from the flexibility subscale also have a good quality of discrimination between .29 and .37. Additionally, something similar occurs in the items selected from the planning and organization subscale, where there is an excellent discrimination of .56 in both of them. 

Items selected from the BRIEF-2 scale were as follows (see link: Confinamiento por COVID-19: impacto psicológico en niños y adolescentes (google.com))

Emotional self-regulation (AR):

1. A su hijo/a le cuesta darse cuenta de cómo su conducta afecta o molesta a los/as demás 

2. Su hijo/a tiene explosiones de ira 

3. Su hijo/a se molesta y enfada con mucha facilidad

4. Su hijo/a reacciona más intensamente que otros chicos/as de su misma edad ante las situaciones

5. Las rabietas, enfados y/o lloros de su hijo/a son intensos, pero ceden repentinamente

6. Su hijo/a tiene frecuentes cambios de humor a lo largo del día

Cognitive Flexibility (FC):

7. Su hijo/a se resiste o le cuesta aceptar maneras alternativas de resolver un problema relacionado con las tareas del colegio, de casa, etc.

8. A su hijo/a le cuesta acostumbrarse a situaciones nuevas (cambios de rutina, horarios, lugares de trabajo, etc.)

9. Su hijo/a muestra dificultades para salir de un tema o de una actividad en la que se quede “enganchado/a"

10. Su hijo/a tiene problemas para cambiar de una actividad a otra

Planning and organization (PO):

11. Su hijo/a hace las tareas escolares y no escolares sin planificarlas previamente

12. A su hijo/a le cuesta organizar actividades con sus amigos (videoconferencias, juegos on-line, etc.)

Point 4: It should be better explained how the association between levels of state anxiety, executive functions and the online teaching model took place;

Thank you for your suggestions, we have revised both the introduction, discussion and conclusions to make them clearer (see sections 1, 4 y 5: introduction, discussion and conclusions)

Did you assess anxiety levels and executive functions even in the period preceding the covid?

In relation to this last question, it is worth mentioning that since the scores prior to the COVID-19 were not available due to the obviousness of the unexpected situation, we proceeded to compare the means found in each of the variables analysed with the means of the scales published for each instrument used, as well as the studies of their psychometric properties. The results obtained are very interesting, as can be seen in section 3. 

Point 5: In the discussion you said that "These studies do not specifically take into account the most vulnerable population that will also constitute the future of our society, a group to which particular importance must be attributed, namely children and adolescents" but there are studies that evaluate the impact of the pandemic on children, for example in the study by Sergi et al. The "Autism, Therapy and COVID-19" group is made up of children with ASD, so it would be better to expand the literature.

The observation highlighted in the discussion refers to the fact that the number of studies found on the effects of the pandemic on the mental health of adults has been higher than what indicates research findings in children and adolescents, especially in the non-pathological population. As you point out, there are more and more colleagues who are carrying out very interesting studies on the effects of the pandemic on the mental health of our normotypical and neurodevelopmentally disordered children and adolescents. We shall now expand the bibliography and qualify the expression so that it is not misleading (see section 4)

Round 2

Reviewer 1 Report

Thank you to the authors for incorporating my comments. I would like to comment once again on one of the points mentioned.

Point 3: The abstract states that "the group aged 11 to 18 years showed significantly higher state anxiety (p = 0.041) than the group aged 6 to 10 years". The p-level value in this case is borderline and due to the number of respondents, statistical significance is expected. At this point, I consider it appropriate to add effect size in addition to statistical significance. The value of "d" can be found in the chapter "results" and how the authors themselves add "The calculated effect size is small". This mention completely reverses the meaning of the text from the abstract.

The authors correctly added the value of material significance to the abstract. However, this value is d = -0.113. And according to Cohen (1988, p. 25), this is a small effect. So it is still true that the sentence: „group aged 11 to 18 years showed methodologically higher state anxiety (p=0.041; 27 d=-0.113) than the group aged 6 to 10 years“ it doesn't make sense to me. I would rather talk about the fact that the differences are statistically significant. However, the effect size is negligible and almost unusable for practice.

Cohen, J. 1988. Statistical Power Analysis for the Behavioral Science (2nd ed.). Hillsdale (NJ): Erlbaum.

In all other cases, the authors have satisfactorily incorporated my comments.

Author Response

Point 3: The abstract states that "the group aged 11 to 18 years showed significantly higher state anxiety (p = 0.041) than the group aged 6 to 10 years". The p-level value in this case is borderline and due to the number of respondents, statistical significance is expected. At this point, I consider it appropriate to add effect size in addition to statistical significance. The value of "d" can be found in the chapter "results" and how the authors themselves add "The calculated effect size is small". This mention completely reverses the meaning of the text from the abstract.

The authors correctly added the value of material significance to the abstract. However, this value is d = -0.113. And according to Cohen (1988, p. 25), this is a small effect. So it is still true that the sentence: „group aged 11 to 18 years showed methodologically higher state anxiety (p=0.041; 27 d=-0.113) than the group aged 6 to 10 years“ it doesn't make sense to me. I would rather talk about the fact that the differences are statistically significant. However, the effect size is negligible and almost unusable for practice.

We absolutely agree with you. Despite the methodological significance of the test, these results are not clinically relevance. We have changed the abstract again (lines 26-28), as well as the results section (lines 305-307). We hope we have modified the text accurately according to your comment.